# The Chicken cGAS–STING Pathway Exerts Interferon-Independent Antiviral Function via Cell Apoptosis

**DOI:** 10.3390/ani13162573

**Published:** 2023-08-09

**Authors:** Sen Jiang, Mengjia Lv, Desheng Zhang, Qi Cao, Nengwen Xia, Jia Luo, Wanglong Zheng, Nanhua Chen, François Meurens, Jianzhong Zhu

**Affiliations:** 1Comparative Medicine Research Institute, Yangzhou University, Yangzhou 225009, China; dx120200161@stu.yzu.edu.cn (S.J.); mx120220967@stu.yzu.edu.cn (M.L.); mz120211564@stu.yzu.edu.cn (D.Z.); mz120201467@stu.yzu.edu.cn (Q.C.); dx120220180@stu.yzu.edu.cn (N.X.); dx120210176@stu.yzu.edu.cn (J.L.); 007297@yzu.edu.cn (W.Z.); hnchen@yzu.edu.cn (N.C.); 2College of Veterinary Medicine, Yangzhou University, Yangzhou 225009, China; 3Joint International Research Laboratory of Agriculture and Agri-Product Safety, Yangzhou 225009, China; 4Jiangsu Co-Innovation Center for Prevention and Control of Important Animal Infectious Diseases and Zoonoses, Yangzhou University, Yangzhou 225009, China; 5Swine and Poultry Infectious Diseases Research Center, Faculty of Veterinary Medicine, University of Montreal, St. Hyacinthe, QC J2S 2M2, Canada; francois.meurens@inra.fr; 6Department of Veterinary Microbiology and Immunology, Western College of Veterinary Medicine, University of Saskatchewan, Saskatoon, SK S7N 5E2, Canada

**Keywords:** cGAS, STING, chicken, interferon, apoptosis, autophagy, antiviral

## Abstract

**Simple Summary:**

The Interferon-independent antiviral function of stimulator of Interferon genes has recently received attention. However, the nature of this Interferon-independent antiviral function is not yet clear, although autophagy induced by stimulator of Interferon genes has been shown to mediate the antiviral activity of stimulator of Interferon genes. In birds, research is still nonexistent. Here, we first validated the antiviral activity of chicken stimulator of Interferon genes and determined the Interferon-independent antiviral activity of chicken stimulator of Interferon genes. Furthermore, it was found that autophagy induced by chicken stimulator of Interferon genes did not affect its antiviral activity. Finally, apoptosis induced by chicken stimulator of Interferon genes was involved in its antiviral process. Our results indicate that chicken stimulator of Interferon genes induces multiple signaling activities, which may be correlated during the antiviral process. This study demonstrates the complexity of the antiviral activity of chicken stimulator of Interferon genes and provides a unique approach for the study of antiviral activity other than that of Interferons.

**Abstract:**

It has been recently recognized that the DNA sensing innate immune cGAS-STING pathway exerts an IFN-independent antiviral function; however, whether and how chicken STING (chSTING) exerts such an IFN-independent antiviral activity is still unknown. Here, we showed that chSTING exerts an antiviral activity in HEK293 cells and chicken cells, independent of IFN production. chSTING was able to trigger cell apoptosis and autophagy independently of IFN, and the apoptosis inhibitors, rather than autophagy inhibitors, could antagonize the antiviral function of chSTING, suggesting the involvement of apoptosis in IFN-independent antiviral function. In addition, chSTING lost its antiviral function in IRF7-knockout chicken macrophages, indicating that IRF7 is not only essential for the production of IFN, but also participates in the other activities of chSTING, such as the apoptosis. Collectively, our results showed that chSTING exerts an antiviral function independent of IFN, likely via apoptosis.

## 1. Introduction

Innate immunity is the first line of defense against pathogenic microorganisms. It recognizes pathogen-associated molecular patterns (PAMPs) and damage-associated molecular patterns (DAMPs) via its encoded pattern recognition receptors (PRRs) [1,2]. PRRs encompass membrane-bound Toll-like receptors (TLRs) and C-type lectin-like receptors (CLRs), cytosolic RIG-I-like receptors (RLRs), NOD-like receptors (NLRs) and cytosolic DNA receptors (CDRs) [1]. Upon activation by PAMPs or DAMPs, PRRs trigger transcription-dependent or -independent cell events, and subsequently drive the expressions of type I interferons (IFNs), pro-inflammatory cytokines or protease activation, orchestrating innate immune responses [2].

Innate immune DNA receptor consist of TLR9 and cytosolic DNA receptors (CDRs) [3]. At present, cytosolic DNA receptors include AIM2, IFI16, LRRFIP1, DHX9, DHX36, DDX41, Ku70, DNA-PK, MRE11, cGAS, STING and Rad50 [3]. Among all, CDRs, the cGAS-STING signaling pathway has been extensively studied as an important host antiviral machinery [4]. The cyclic GMP-AMP synthase (cGAS), as a nucleotidyltransferase (NTase), can be activated by DNA and synthesize cyclic guanosine monophosphate-adenosine monophosphate (2′3′-cGAMP) using ATP and GTP [5]. The 2′3′-cGAMP and bacterial cyclic dinucleotide (CDN), as the second messages, activate STING [6]. The activated STING recruits TBK1 through the TBM motif (D/E)xPxPLR(S/T)D, and TBK1 in turn phosphorylates STING on conserved serine in the pLxIS motif [7]. The phosphorylated STING recruits transcription factor IRF3 through its pLxIS motif, and the recruited IRF3 is phosphorylated by the adjacent TBK1 [8]. Then, the phosphorylated IRF3 enters into the nucleus to initiate transcription and induce the expressions of IFN and IFN-stimulated genes (ISGs), inducing a robust antiviral state [9,10]. STING also activates the transcription factor NF-κB and the production of various pro-inflammatory cytokines [11], further strengthening antiviral activity.

In addition to IRF3- and NF-κB-mediated gene transcriptions, STING also triggers cell autophagy and apoptosis [12,13]. Recent studies have shown that STING can cause canonical autophagy and non-canonical autophagy [12]. STING was shown to induce endoplasmic reticulum (ER) stress and unfolded protein response (UPR) through a unique UPR motif in the cyclic dinucleotide binding (CBD) domain, which negatively regulates the AKT/TSC/mTOR pathway to enhance canonical autophagy [14]. In addition, STING is also able to induce non-canonical autophagy by directly interacting with microtubule-associated protein light chain 3 (LC3) through its LC3 interacting regions (LIRs) [15], and the ER-Golgi intermediate compartment (ERGIC) is used as the membrane source of LC3 lipidation and autophagy formation [15,16,17]. Apoptosis, also known as the non-inflammatory type of programmed cell death, is one of the most widely studied cell death pathways [13,18]. The mechanism of STING inducing apoptosis is not clear, but most studies point to ER stress [19,20,21]. Additionally, STING may induce apoptosis through downstream IRF3 or NF-κB, where the phosphorylation of IRF3 and formation of the IRF3-Bax complex occur [19,20,22].

The IFN-independent activity of STING has been widely appreciated in the last years [23]. Recently, we demonstrated that porcine STING (pSTING) can play its antiviral function independently of both IFN and autophagy, while relating to apoptosis [24]. Our previous study also found that chicken STING (chSTING) can induce broad-spectrum antiviral activity, but the IFN-independent antiviral activity of chSTING has not been elucidated [25]. In this study, we investigated the resistance of chSTING against a variety of viruses and showed that chSTING can induce a robust IFN-independent antiviral activity. We further found that apoptosis, rather than autophagy, participates in the IFN-independent antiviral activity of chSTING.

## 2. Materials and Methods

### 2.1. Reagents and Viruses

TRIpure reagent (34183AX) was purchased from Aidlab (Beijing, China). HiPure DNA Mini Kit (D3121-02) was obtained from Guangzhou Magen Biotechnology Co., Ltd. (Guangzhou, China). HiScript^®^ 1st Strand cDNA Synthesis Kit (R312-02), ChamQ Universal SYBR qPCR Master Mix (Q711), 2×Taq Master Mix (Dye plus) (P222-01) and Annexin V-FITC/PI Apoptosis Detection Kit (A211-01) were all from Vazyme Biotech Co., Ltd. (Nanjing, China). The double-luciferase reporter assay kit (FR201-01-V2) was bought from TransGen Biotech (Beijing, China). Further, 2′3′-cGAMP (9523-44-01) was bought from InvivoGen (Hong Kong, China). The apoptosis inhibitors, Ac-DEVD-CHO (a caspase 3 inhibitor) (C1206), Z-VAD-FMK (a pan caspase inhibitor) (C1202) and autophagy inhibitor ammonium chloride (NH4Cl, an autophagic lysosome inhibitor) (ST2030), were from Beyotime Biotechnology (Shanghai, China). Another autophagy inhibitor, 3-methyladenine (3-MA, a PI3K inhibitor) (HY-19312), was purchased from MedChemExpress (Shanghai, China). The Herpes Simplex Virus 1 (HSV1)-GFP (KOS strain), Vesicular Stomatitis Virus (VSV)-GFP (Indiana HR strain), Sendai Virus (SeV)-GFP, Newcastle Disease Virus (NDV)-RFP and Vaccinia Viruses (VACV and SMV) were all as described and used previously [25]. These viruses are all cultured routinely in cell lines in the DMEM medium in the laboratory.

### 2.2. Antibodies

The mouse anti-actin monoclonal antibody (mAb) and mouse anti-GFP mAb were acquired from Transgen Biotech (Beijing, China). The rabbit phosphorylated p62 (p-p62) polyclonal (p) Ab (18420-1-AP) was purchased from ProteinTech (Wuhan, China). The mouse GAPDH mAb was from ABclonal (Wuhan, China). The rabbit TBK1 mAb (3504S), rabbit phosphorylated-TBK1 (p-TBK1, Ser172) mAb (5483S), IRF3 mAb (11904S), HA mAb (3724) and rabbit LC3B (D11) XP mAb (3868) were all acquired from Cell Signaling Technology (Boston, MA, USA). The rabbit phospho-IRF3 (p-IRF3, Ser385) pAb (MA5-14947) was purchased from Thermo Fisher Scientific (Sunnyvale, CA, USA). Anti-RFP mAb HRP-DirecT (M204-7) was purchased from MBL Beijing Biotech Co., Ltd. (Beijing, China). Rabbit ISG56 pAb was homemade and stored at our lab. HRP goat anti-rabbit IgG (H+L) highly cross-adsorbed secondary antibody and goat anti-mouse IgG (H+L) highly cross-adsorbed secondary antibody were obtained from Sangon Biotech (Shanghai, China).

### 2.3. Cells and Cell Transfections

HEK-293T cells (ATCC Cat# CRL-3216), Vero cells (ATCC Cat# CCL-81) and DF-1 cells (ATCC Cat# CRL-12203) were maintained in the Dulbecco modified Eagle medium (DMEM, Hyclone Laboratories, Logan, UT, USA), supplemented with 10% fetal bovine serum (FBS, Gibco, Billings, MT, USA) and 1% penicillin/streptomycin, and maintained at 37 °C with 5% CO_2_. Chicken HD11 macrophages (BCRJ Cat# 0099) were cultured in the Roswell Park Memorial Institute medium (RPMI) 1640 medium (Hyclone Laboratories, USA), supplemented with 10% FBS and 1% penicillin/streptomycin, and maintained at 37 °C with 5% CO_2_. The gene knockout HD11 cells (STING^−/−^ cells and IRF7^+/−^ cells) were as described previously [26]. Transfection was performed by using Lipofectamine 2000 or TransIT-LT1 (Thermo Fisher Scientific), following the manufacturer’s instructions.

### 2.4. Molecular Cloning and Gene Mutations

The HA-tagged pcDNA DEST plasmids of chicken cGAS (chcGAS) and chicken STING (chSTING) were made as in our previous studies [25,26]. The mutation PCR primers of chSTING were designed using the QuickChange Primer Design online tool (https://www.agilent.com, accessed on 1 October 2020), shown in Appendix A. The mutation PCR was performed with KOD plus neo polymerase (Toyobo, Shanghai, China) and pcDNA-chSTING-2HA as the template. The PCR products were transformed into competent DMT *Escherichia coli* after Dpn I digestion, and the resultant mutants were sequence, confirmed by DNA sequencing.

### 2.5. Promoter-Driven Luciferase Reporter Gene Assay

In 96-well plates (2–3 × 10^4^ cells/well), 293T cells were co-transfected using Lipofectamine 2000 with ISRE or ELAM (NF-κB) firefly luciferase (Fluc) reporter (10 ng/well) and β-actin Renilla luciferase (Rluc) reporter (0.2 ng/well), together with the indicated plasmids or vector control (5–40 ng/well). DF-1 cells were similarly transfected with chicken IFNβ Fluc using TransIT-LT1 transfection reagent (MIR 2300). The total DNA per well was normalized to 50 ng by adding corresponding empty vectors. Twenty-four hours after transfection, the cells were lysed, and the Fluc and Rluc activities were measured successively by the double-luciferase reporter assay kit. The results were expressed as the fold induction of IFNβ, ISRE or ELAM (NF-κB)-Fluc compared with those of vector controls after Fluc normalization by the corresponding Rluc.

### 2.6. RNA and DNA Extraction, Reverse Transcription and Quantitative PCR (qPCR)

Total RNA was extracted using the TRIpure reagent, and total DNA was extracted using a DNA extraction kit following the instructions. The extracted RNA was reverse-transcribed into cDNA using a HiScript 1st strand cDNA synthesis kit, and then the target gene expressions were measured using quantitative PCR with a SYBR qPCR master Mix using StepOne Plus equipment (Applied Biosystems, Waltham, MA, USA). The qPCR program included a denaturation step at 94 °C for 30 s, followed by 40 cycles at 94 °C for 5 s and 60 °C for 30 s. The relative mRNA levels were normalized to RPL32 or GAPDH mRNA levels, and calculated using the 2^−ΔΔCT^ method. The efficiencies of the qPCR assays were always at 90 to 110%, requiring use of the 2^−ΔΔCT^ method. The sequences of qPCR primers used are shown in Appendix A.

### 2.7. Western Blotting Analysis

Whole-cell proteins were extracted from cells lysed using the radioimmunoprecipitation assay (RIPA) lysis buffer (50 mM Tris pH 7.2, 150 mM NaCl, 1% sodium deoxycholate, 1% Triton X-100). The protein lysate samples were 3:1 mixed with 4× loading buffer and boiled for 5–10 min. After centrifugation, the lysate supernatants were run by SDS-PAGE, and then the proteins in gel were transferred to the polyvinylidene fluoride (PVDF) membrane. The membrane was incubated with 5% skim milk TBST at room temperature (RT) for 1 h, probed with the indicated primary antibodies at 4 °C overnight, washed, and then incubated with secondary antibodies for 1 h at RT. The protein signals were detected via the electrochemiluminescence (ECL) detection substrate and imaging system (Tanon, Shanghai, China).

### 2.8. Plaque Assay

Vero cells were seeded into 24-well plates (3 × 10^5^ cells/well). Once the cells were grown into a monolayer, cells were infected with the tenfold serially diluted cell supernatants from HSV1-, VSV- or SeV-infected cells for 2 h. Then, the infected cells were washed with PBS and overlaid by immobilizing a medium of a 1:1 mixture of warmed 2× DMEM with 4% FBS and a stock solution of heated 1.6% low-melting agarose (Sigma-Aldrich, St. Louis, MO, USA). Plague formations took 4 days for HSV1, and 2 days for VSV and SeV, respectively. Upon completion, the immobilizing medium was discarded by carefully tipping and the cells were fixed and stained with a crystal violet cell colony staining solution (0.05% *w*/*v* crystal violet, 1% formaldehyde, 1× PBS and 1% methanol) for 1 h at RT. Finally, cells were gently washed with tap water until the clear plaques appeared. The plaques were counted, and photos were taken.

### 2.9. Viral Fluorescence Imaging

The viral fluorescence was imaged using the OLYMPUS IX53 fluorescence microscope, with the objective lens parameter of 10 × 0.30 ph1. For red fluorescent protein (RFP) imaging, the excitation wavelength and emission wavelength were 532 nm (green light) and 588 nm, respectively. For green fluorescent protein (GFP) imaging, the excitation wavelength and emission wavelength were 488 nm (blue light) and 507 nm, respectively. The microscopic image processing software used was OLYMPUS cellsens Entry 2.2 (Build 17989) (https://www.olympus-sis.com). The same set of samples were exposed for the same time, and three typical images were collected for each sample.

### 2.10. Flow Cytometry for Apoptosis Detection

The level of cell apoptosis was examined using the Annexin V fluorescein isothiocyanate (FITC)/propidium iodide (PI) Apoptosis Detection Kit. Briefly, the treated cells were harvested via trypsin digestion and washed with the binding buffer, and then re-suspended in the binding buffer. The staining solution of Annexin V-FITC and PI were added successively. The cells were incubated with fluorescein dyes at RT for 15 min in the dark, and the stained cells were immediately detected using flow cytometry. About 10,000 cell events were collected for analysis of FITC and PI signals detected by channels FL1 and FL3, respectively. The FITC and PI signals were processed using the FlowJo software v10.9 (https://www.flowjo.com), and presented as dot plots. The flow cytometry analyzer used was CytoFLEX S (Beckman Coulter, Brea, CA, USA), equipped with four laser beams (488 nm laser, 638 nm laser, 405 nm laser, 561 nm laser).

### 2.11. Statistical Analysis

All the results were the representatives of two or three similar experiments. The data in bar graphs were presented as the mean ± standard deviation (SD) and analyzed using GraphPad Prism 8.0. The normality of data distribution was tested using the Shapiro-Wilk test. Student’s *t*-test and ANOVA were used for statistical analysis, and *p* < 0.05 was considered statistically significant. Statistical signs are indicated as follows: *, *p* < 0.05; **, *p* < 0.01 and ns, not significant.

## 3. Results

### 3.1. chSTING Exerts an IFN-Independent Antiviral Activity in Mammalian 293T Cells

Our previous study showed that chSTING could not induce IFN in mammalian cells; however, substituting its pLxVS motif with a porcine pLxIS motif enables the induction of IFN in mammalian 293T cells by chSTING [26]. We examined the antiviral activity of chSTING WT and its pLxVS sub-mutant in 293T cells (Figure 1). Despite the difference in IFN induction, both chSTING and its pLxVS sub-mutant exhibited antiviral activity against a DNA virus (HSV1) (Figure 1A) and two RNA viruses (VSV and SeV) (Figure 1B,C), as evidenced by GFP fluorescence, viral GFP blotting and viral plaque assay, when co-transfected with chcGAS. These results indicated the IFN-independent antiviral activity of chSTING in mammalian 293T cells.

To further determine the IFN-independent antiviral effect of chSTING in 293T cells, two IFN-deficient chSTING mutants (pLxVS sub S366A and ∆CTT) were constructed from the IFN-capable chSTING pLxVS sub. The serine 366 of chSTING is the TBK1 phosphorylation site, which is critical for IRF7 recruitment and downstream IFN induction. The chSTING CTT contains both TBK1 recruitment motif (TBM) and IRF7 recruitment motif (pLxVS), and is thus essential for IFN induction (Figure 2A). In the promoter assay, neither chSTING pLxVS sub S366A nor chSTING ∆CTT had the ISRE promoter activity as expected (Figure 2B), but both still had NF-κB promoter activity, despite the variation (Figure 2C). Consistently, in Western blotting, both chSTING pLxVS sub S366A and chSTING ∆CTT could not activate IRF3 phosphorylation and downstream ISG56 production, with the former (S366A) but not the latter one able to activate TBK1 phosphorylation (Figure 2D). An RT-qPCR assay showed that chSTING pLxVS sub S366A and chSTING ∆CTT could not induce IFN-β, ISG56 and ISG60, but kept the ability to induce IL-8 and TNF-α in 293T cells (Figure 2E). Upon co-transfection with chcGAS, two IFN-deficient chSTING mutants still resisted effectively against a virus infections with HSV1 (Figure 2F and Appendix A) and VSV (Figure 2G and Appendix A). Together, the results clearly demonstrated that chSTING exerts an IFN-independent antiviral activity in mammalian 293T cells.

### 3.2. chSTING Exhibits a Robust IFN-Independent Antiviral Function in Chicken Cells

Considering the physiological relevance, we selected chicken cells (chicken fibroblast cell line DF-1) for further research. Unlike 293T, chSTING alone has significant signaling activity when transfected into DF-1 alone, and its activity does not increase further after co-transfection with cGAS [26]. Accordingly, ectopic chSTING in DF-1 cells exhibited significant antiviral responses to NDV, vaccinia viruses SMV and VACV, whereas co-expression with chcGAS did not enhance such antiviral responses (Figure 3A). As expected, chcGAS alone had no observable antiviral effect as evidenced by RFP fluorescence and RFP immune blotting for NDV, and by qPCR for VACV and SMV (Figure 3B).

In DF-1 cells, different chSTING mutants, including chSTING pLxVS, chSTING S366A and chSTING ∆CTT, were examined for their IFN activity (Figure 2A and Figure 3C,D). The chIFNβ promoter assay showed that, compared with the chSTING WT, chSTING pLxVS sub had poor IFNβ activity due to species specificity, while chSTING S366A and ∆CTT had no IFNβ activity (Figure 3C). Similarly, RT-qPCR showed that, relative to chSTING WT, chSTING pLxVS sub had low inductions of IFNβ and OASL, while chSTING S366A and ∆CTT had no induction of IFNβ and OASL (Figure 3D). On the other hand, chSTING WT, pLxIS sub and S366A were all able to induce IL-8, but chSTING ∆CTT could not (Figure 3D).

As for antiviral function, compared with chSTING WT, the chSTING pLxVS sub and chSTING S366A exhibited robust anti-NDV activity, whereas chSITNG ∆CTT also showed anti-NDV activity despite it being to a lower degree (Figure 3E). On the other hand, chSTING WT and all the chSTING mutants (pLxVS sub, S366A and ∆CTT) exhibited significant antiviral activity against VACV and SMV (Figure 3F). Notably, the above results could be reproduced in the chicken macrophage HD11 cell line (Figure 3G). This result suggested that chSTING exhibits an IFN-independent antiviral activity in chicken cells as well, and indicated that chSTING possesses other antiviral mechanisms in addition to IFN.

### 3.3. Apoptosis Rather Than Autophagy Is Involved in the Antiviral Function of chSTING

Since it was determined that chSTING can play its antiviral role without IFN in different types of cells, the inherent autophagy and apoptosis activity of chSTING and its three mutants were tested in 293T cells (Figure 4). The Western blotting showed that chSTING and all three mutants (pLxVS sub, pLxVS sub S366A and ∆CTT) upon chcGAS stimulation induced LC3 lipidation (LC3-II) and p62 phosphorylation (p-p62), indicative of the autophagy occurrence (Figure 4A). Flow cytometry showed that chSTING and all chSTING mutants were able to induce significant apoptosis (Figure 4B). Then, the statistical analysis was performed, and it was found that all chSTING mutants, including IFN-deficient mutants, could induce early and late apoptosis (Figure 4C). These results suggested that chSTING-induced apoptosis and autophagy were both independent of IFN.

In order to determine whether the apoptosis and/or autophagy of chSTING participates in the antiviral activity of chSTING, two autophagy inhibitors (3-MA and NH_4_Cl) (Figure 4D) and two apoptosis inhibitors (DEVD and VAD) (Figure 4E) were applied to antiviral experiments in chicken HD11 cells. STING agonist 2′3′-cGAMP triggered an obvious anti-NDV response in HD11 cells (Figure 4D,E). The two autophagy inhibitors did not significantly affect the antiviral effect of 2′3′-cGAMP stimulation (Figure 4D), whereas the two apoptosis inhibitors could reverse the antiviral response activated by 2′3′-cGAMP to some extent, compared with DMSO control group (Figure 4E). The above results suggested that apoptosis, rather than autophagy, induced by chSTING is likely involved in the IFN-independent antiviral function of chSTING, which was consistent with the results of pSTING we previously studied [24].

### 3.4. IRF7 Is Required for STING Antiviral Function in Chicken HD11 Cells

In order to further investigate the antiviral function of chSTING, STING^−/−^ HD11 cells were used for NDV infection experiment. Compared with the WT HD11, STING^−/−^ HD11 had a much higher NDV replication level (Figure 5A). The STING agonist, 2′3′-cGAMP, suppressed the NDV replication potently in WT HD11 cells, but lost the antiviral activity in STING^−/−^ HD11 cells (Figure 5A). The results confirmed and reinforced the critical role of chSTING in antiviral innate immunity.

Chicken IRF7 (chIRF7) is a transcription factor that mediates IFN induction by chSTING; therefore, the IRF7^+/−^ HD11 cells were also examined for NDV infection. Surprisingly, IRF7^+/−^ HD11 cells had a much higher NDV replication level compared with the WT HD11 cells and the STING agonist, 2′3′-cGAMP, lost its antiviral activity in IRF7^+/−^ HD11 cells (Figure 5B). These results indicated that chIRF7 plays a significant role in the process of anti-NDV replication likely via both IFN-dependent and IFN-independent ways. To explore whether chIRF7 is involved in chSTING-induced apoptosis, we analyzed the cell apoptosis in IRF7-knockout cells. The results showed that the cleaved caspase 3 in IRF7^+/−^ HD11 cells was substantially decreased with or without cGAMP stimulation compared with those in WT HD11 cells (Figure 5C). Furthermore, when the IRF7^+/−^ HD11 cells were treated with two apoptosis inhibitors, no further increase in NDV replication was observed (Figure 5D). These results implicated that chIRF7 is directly involved in cell apoptosis.

## 4. Discussion

STING, as the signaling adaptor downstream of cGAS, plays a key role in antiviral innate immunity [4]. In recent years, chSTING has also been reported to resist Marek’s disease virus (MDV, member of the Herpesviridae family) and fowl adenovirus [27,28,29], both avian DNA viruses. Later on, we showed that chSTING also resists the replications of a broad spectrum of viruses including DNA virus, RNA virus and retrovirus [25]. The broad-spectrum antiviral activity of chSTING has also been further confirmed in this study: chSTING could reduce the replications of HSV1, VSV and SeV in mammalian 293T cells (Figure 1 and Figure 2), and suppress the replications of NDV, SMV and VACV in chicken cells (Figure 3, Figure 4 and Figure 5).

Despite that, the IFN-independent antiviral activity of STING has been appreciated [23,30,31]. In this study, we first demonstrated the IFN-independent antiviral activity of chSTING in both mammalian and chicken cells (Figure 1, Figure 2 and Figure 3). It is known that STING can not only induce IFN, but also activate NF-κB, autophagy and apoptosis [12,13,32]. First, chSTING pLxVS sub S366A and chSTING ∆CTT have differential NF-κB signaling (Figure 2C), but the antiviral activity of these two chSTING mutants are similar (Figure 2F,G). Thus, it indicates that NF-κB does not affect apparently the antiviral activity of chSTING. Second, although some studies pointed out that STING-induced autophagy was involved in the antiviral process [16,31,33], we did not find any involvement of autophagy in the antiviral activity of chSTING (Figure 4). Third, the two kinds of apoptosis inhibitors could reverse STING agonist 2′3′-cGAMP triggered antiviral activity, suggesting that apoptosis is likely involved in the IFN-independent antiviral process of chSTING (Figure 4). The results are consistent with those obtained in our previous study on porcine STING (pSTING), reflecting the conservation of STING during evolution.

Cell apoptosis is the key to control and eliminate viral infection [34]. When IFN cannot control the influenza virus, the host can induce cell apoptosis to inhibit viral replication [35]. Many viruses have evolved a variety of apoptosis-inhibiting proteins to evade cell apoptosis at an early stage of infections and thus contribute to their own replications, including porcine respiratory and reproductive syndrome virus (PRRSV) [36], human cytomegalovirus (HCMV) [37], α-herpesvirus [38] and ectromelia virus [39]. Thus, apoptosis induced by chSITNG serving as the IFN-independent antiviral machinery might be counteracted by chicken viruses, as well for immune evasion.

Here, we found that chIRF7 is required for STING antiviral response to NDV in chicken HD11 cells (Figure 5). Several DNA viruses including HSV1 and Vaccinia Viruses SMV and VACV did not replicate well in HD11 cells, and other relevant viruses need to be tested for this interesting phenomenon. In fact, in our previous pSTING work, we found that IRF3 is also required for STING antiviral activity [24]. The similar phenotypes seem contradictory to the IFN-independent antiviral activity of STING. However, it will be reconciled if the IRF3/IRF7, required for IFN induction by STING, also participated in the IFN-independent activity of STING, such as apoptosis. Although we showed here that chIRF7 is likely involved in chSTING triggered cell apoptosis directly, the hypothesis needs to be confirmed in the future. In addition, the mechanism of apoptosis by STING is not clear currently. In particular, the responsible sites and/or molecular switch of STING for apoptosis need to be identified and clarified.

## 5. Conclusions

In conclusion, our study first demonstrated the IFN-independent antiviral activity of chSTING and found that the apoptosis triggered by chSTING is involved in the antiviral function and likely the IFN-independent antiviral function of chSTING. Our study further suggested that dissecting the molecular mechanism of STING apoptosis is very meaningful in antiviral research.

## Figures and Tables

**Figure 1 animals-13-02573-f001:**
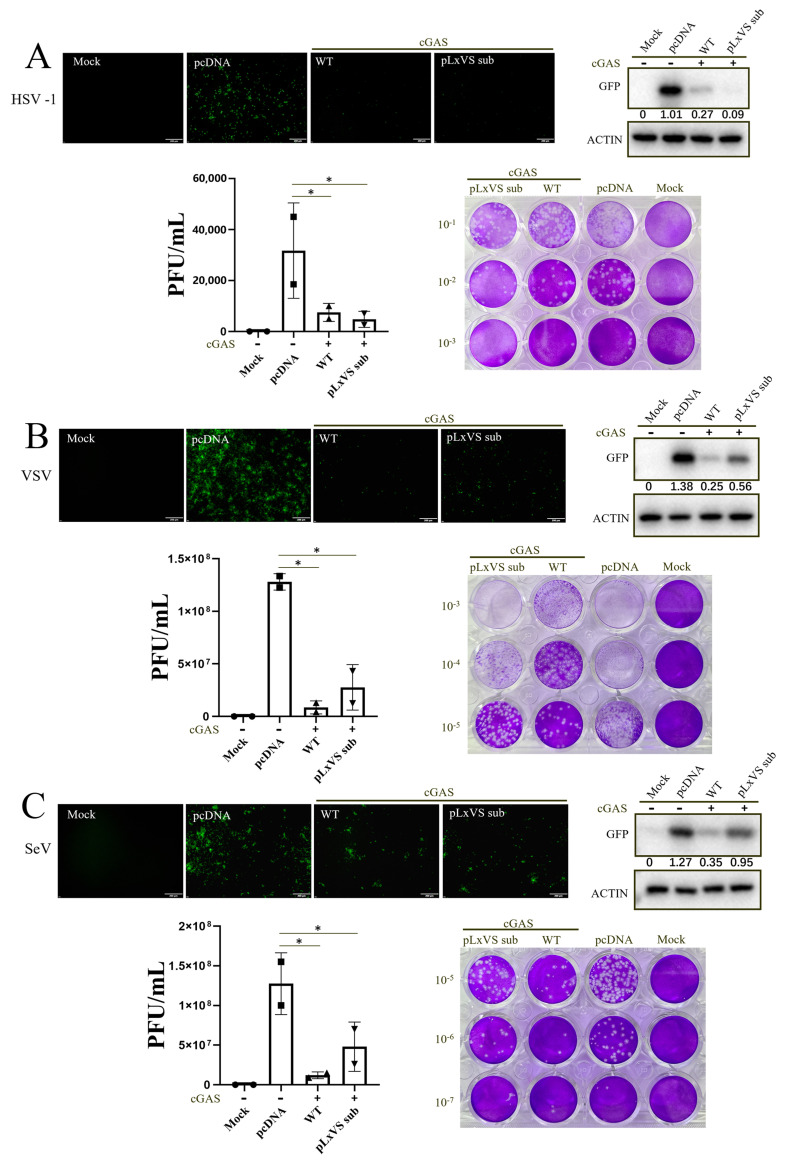
chSTING does not rely on IFN to play its anti-viral functions in 293T. (**A**) The 293T cells in a 24-well plate (1.5 × 10^5^ cells/well) were co-transfected with chcGAS (0.25 μg) combined, with chSTING WT and pLxVS sub (0.25 μg each) for 24 h, with pcDNA3.1 empty vector transfection and mock as controls. The transfected cells were infected with HSV1-GFP (0.1 MOI) for another 24 h. The viral GFP fluorescence was observed by microscopy. The expressions of viral GFP protein were detected by Western blotting. HSV1 plaques and the numbers of plaques were presented. (**B**,**C**) The 293T cells were transfected as in HSV1 infection. The transfected cells were infected with VSV-GFP (0.01 MOI) (**B**) and SeV-GFP (0.01 MOI) (**C**) for another 12 h. The viral GFP fluorescence was observed via microscopy. The expressions of viral GFP protein were detected through Western blotting. The gray values of GFP protein bands in Western blotting were shown below the corresponding bands after normalization of actin expressions. VSV and SeV plaques and the number of plaques were presented. * *p* < 0.05 versus Mock controls.

**Figure 2 animals-13-02573-f002:**
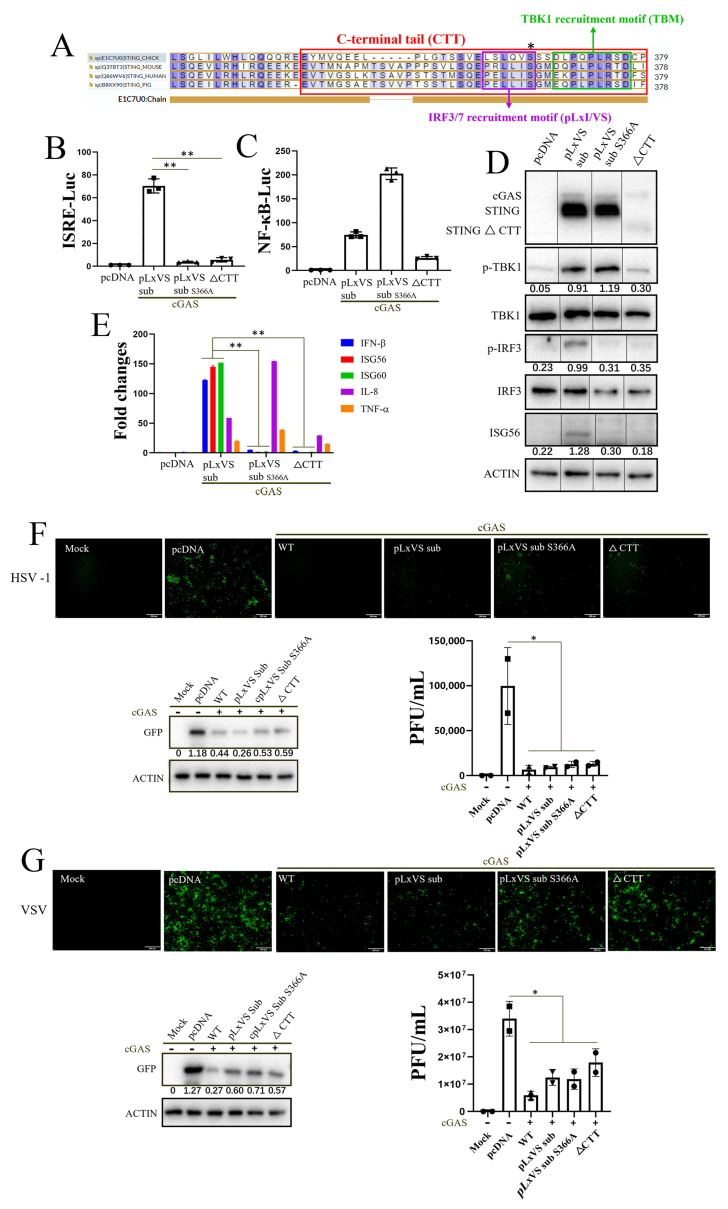
The IFN-deficient chSTING mutants can still resist viral infections in 293T cells. (**A**) The CTT domain (red box), TBM motif (green box), pLxI/VS motif (purple box) and S366 (star sign) of chSTING are predicted based on the sequence alignment. (**B**,**C**) The 293T cells in 96-well plate (3 × 10^4^ cells/well) were transfected with the indicated combinations of chcGAS (20 ng) and chSTING pLxVS sub/pLxVS sub S366A/∆CTT (20 ng each), or pcDNA3.1 vector (40 ng), together with ISRE Fluc (10 ng) (**B**) or NF-κB Fluc (10 ng) (**C**) plus Rluc (0.2 ng). Luciferase activities were measured 24 h after transfection. (**D**) 293T cells grown in a 24-well plate (1.5 × 10^5^ cells/well) were co-transfected with the indicated combinations of chcGAS (0.25 μg) and chSTING pLxVS sub/pLxVS sub S366A/∆CTT (0.25 μg each) or pcDNA3.1 vector (0.5 μg) for 24 h, and the expressions of cGAS-HA, STING-HA, p-TBK1, TBK1, p-IRF3, IRF3 and ISG56 were examined via Western blotting. The protein bands separated by the vertical lines were from non-contiguous portions of one gel. (**E**) 293T cells grown in a 12-well plate (3 × 10^5^ cells/well) were transfected with the indicated combinations of chcGAS (0.5 μg) and chSTING pLxVS sub/pLxVS sub S366A/∆CTT (0.5 μg each) for 48 h. The RNA expressions of hIFN-β, hISG56, hISG60, hIL-8 and hTNF-α were analyzed via RT-qPCR. (**F**,**G**) The 293T cells were transfected with chcGAS (0.25 μg) combined with chSTING WT/pLxVS sub/pLxVS sub S366A/∆CTT (0.25 μg each) for 24 h, with the pcDNA3.1 transfection and mock as controls. The treated cells were infected with HSV1-GFP (0.1 MOI) for 24 h (**F**) and VSV-GFP (0.01 MOI) for 12 h (**G**), followed by detection of viral replications with GFP fluorescence, GFP protein expressions and plaque assay. The gray values of p-TBK1/TBK1, p-IRF3/IRF3, ISG56/actin and GFP/actin are shown below the corresponding bands in Western blotting. * *p* < 0.05, ** *p* < 0.01 versus Mock controls.

**Figure 3 animals-13-02573-f003:**
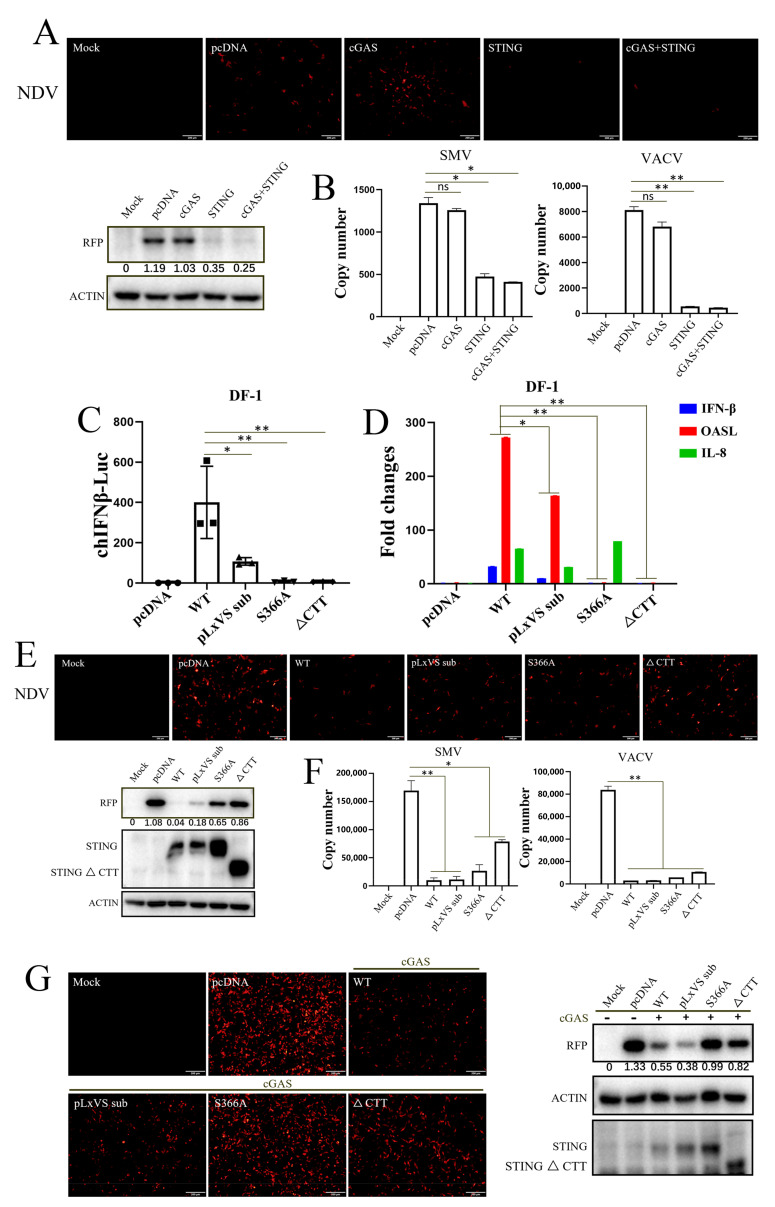
chSTING can exert antiviral function independently of IFN in chicken cells. (**A**) DF-1 cells in a 24-well plate (3 × 10^5^ cells/well) were transfected with chcGAS (0.25 μg), chSTING (0.25 μg), or both (0.25 μg each) for 24 h, with the pcDNA3.1 vector transfection and mock treatment used as controls. The transfected cells were infected with NDV-RFP (0.01 MOI) for 12 h. The red fluorescence of RFP was observed, and the RFP protein was detected via Western blotting. (**B**) DF-1 cells were transfected as in A, and the transfected cells were infected with SMV and VACV (0.01 MOI), respectively, for 48 h, followed by qPCR detection of the viral replication. (**C**) DF-1 cells in a 96-well plate (6 × 10^4^ cells/well) were transfected with chSTING WT/pLxVS sub/S366A/∆CTT (40 ng each), or empty vector (40 ng), together with IFNβ Fluc (10 ng) plus Rluc (0.2 ng) for 24 h, followed by the measure of luciferase activities. (**D**) DF-1 cells grown in a 12-well plate (6 × 10^5^ cells/well) were transfected with chSTING WT/pLxVS sub/S366A/∆CTT (1 μg each) or empty vector (1 μg) for 48 h. The RNA expressions of chIFN-β, chOASL and chIL-8 were analyzed using RT-qPCR. (**E**) DF-1 cells in a 24-well plate (3 × 10^5^ cells/well) were transfected with chSTING WT/pLxVS sub/S366A/∆CTT (0.5 μg each) for 24 h. The transfected cells were infected with NDV-RFP (0.01 MOI) for 12 h, followed by visualization of RFP and immunoblotting analysis of RFP and STING protein expressions. (**F**) DF-1 cells were transfected as in E, and transfected cells were infected with SMV and VACV (0.01 MOI) for 48 h, followed by qPCR detection. (**G**) STING^−/−^ HD11 cells in a 24-well plate (3 × 10^5^ cells/well) were transfected as indicated for 24 h, and infected with NDV for 12 h, followed by visualization of red fluorescence of RFP and immunoblotting analysis of RFP and STING protein expressions. The gray values of RFP/actin were shown below the RFP bands in Western blotting. * *p* < 0.05, ** *p* < 0.01 versus Mock controls. ns, not significant.

**Figure 4 animals-13-02573-f004:**
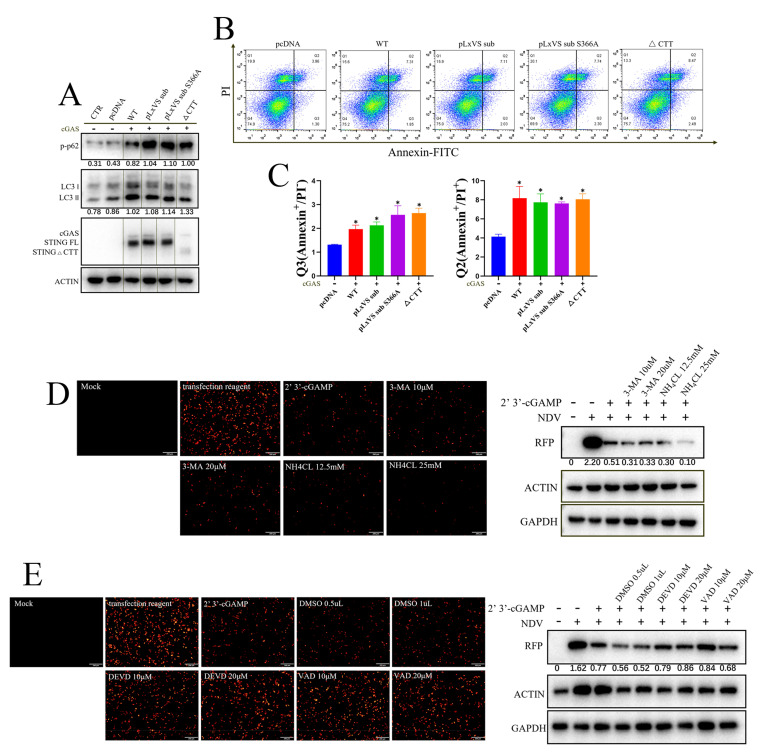
Apoptosis, rather than autophagy, is involved in the antiviral process of chSTING. (**A**) The 293T cells grown in a 24-well plate (1.5 × 10^5^ cells/well) were transfected with the indicated combinations of chcGAS (0.25 μg) and chSTING WT/pLxVS sub/pLxVS sub S366A/∆CTT (0.25 μg each) for 24 h, and the protein expressions were analyzed via Western blotting with the indicated antibodies. The protein bands separated by vertical lines were from non-contiguous portions of one gel. (**B**,**C**) The 293T cells in a 24-well plate (1.5 × 10^5^ cells/well) were transfected with chcGAS (0.25 μg) combined with chSTING WT/pLxVS sub/pLxVS sub S366A/∆CTT, as indicated (0.25 μg each), for 48 h. The apoptosis of the transfected cells was analyzed using Annexin V-FITC/PI staining followed by flow cytometry. The dot plots of cell apoptosis in various samples are presented in B. As indicated bt Q3 (Annexin^+^/PI^−^), the early apoptosis and late apoptosis as indicated by Q2 (Annexin^+^/PI^+^) were plotted as bar graphs in C. (**D**) HD11 cells grown in a 24-well plate (3 × 10^5^ cells/well) were treated with two autophagy inhibitors, 3-MA and NH_4_CL, with the indicated concentrations for 30 min. Cells were washed and stimulated by the transfection of 2′3′-cGAMP (2 μg/mL) for 12 h. The stimulated cells were infected with NDV-RFP (0.01 MOI) for another 12 h, followed by detections of viral RFP fluorescence and RFP protein expression. (**E**) HD11 cells were treated with two apoptosis inhibitors, Ac-DEVD-CHO and Z-VAD-FMK, with the indicated concentrations for 30 min. Cells were washed and stimulated by transfection of 2′3′-cGAMP (2 μg/mL) for 12 h. The stimulated cells were infected with NDV-RFP (0.01 MOI) for another 12 h, followed by the detection of viral replications. The gray values of p-p62/actin, LC3II/LC3I and RFP/GAPDH are shown below the corresponding bands in Western blotting. * *p* < 0.05 versus Mock controls.

**Figure 5 animals-13-02573-f005:**
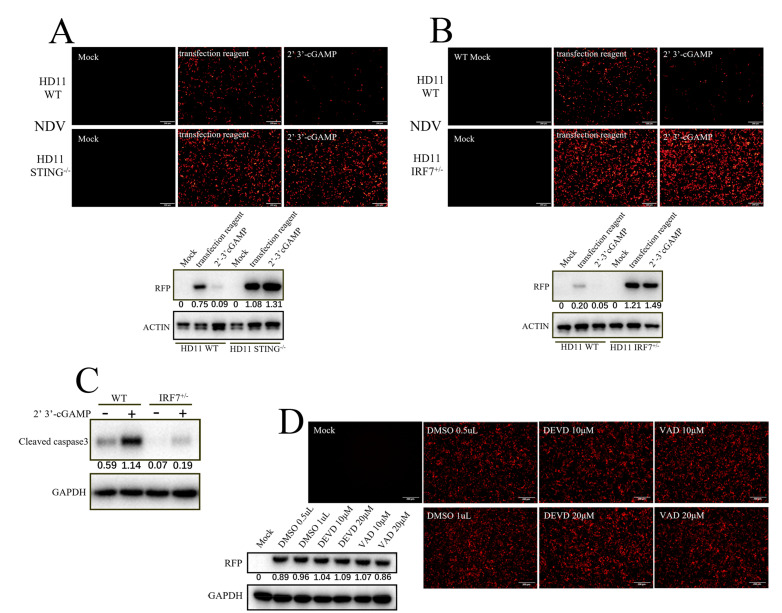
Both chSTING and chIRF7 play an important role in 2′3′-cGAMP-induced antiviral response. (**A**) STING^−/−^ HD11 and WT HD11 cells in a 24-well plate (3 × 10^5^ cells/well) were stimulated with transfection of 2′3′-cGAMP (2 μg/mL) using TransIT-LT1 for 12 h. The stimulated cells were infected with NDV-RFP (0.01 MOI) for 12 h, followed by detection of viral replications with RFP fluorescence and RFP protein expressions. (**B**) IRF7^+/−^ HD11 and WT HD11 cells in a 24-well plate were stimulated with transfection of 2′3′-cGAMP (2 μg/mL) using TransIT-LT1 for 12 h. The stimulated cells were infected with NDV-RFP (0.01 MOI) for 12 h, followed by detection of viral replications with RFP fluorescence and RFP protein expressions. (**C**) IRF7^+/−^ HD11 and WT HD11 cells in a 24-well plate were stimulated with transfection of 2′3′-cGAMP (2 μg/mL) using TransIT-LT1 for 24 h, followed by detection of cleaved caspase 3 protein expressions. (**D**) IRF7^+/−^ HD11 cells in a 24-well plate were treated with two apoptosis inhibitors, Ac-DEVD-CHO and Z-VAD-FMK, with the indicated concentrations for 30 min. The treated cells were infected with NDV-RFP (0.01 MOI) for another 12 h, followed by detection of viral replications.

## Data Availability

No new data created.

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
