# Peer review of "The Chicken cGAS–STING Pathway Exerts Interferon-Independent Antiviral Function via Cell Apoptosis"

_animals, 2023, doi:10.3390/ani13162573_

Round 1

Reviewer 1 Report

This article by Jiang and colleagues investigates the interferon-independent antiviral activities of the chicken STING protein (chSTING). The authors show that chSTING induces apoptosis and autophagy with or without interferon present in response to viral infections. Importantly, they show that it is the apoptotic function of chSTING that antagonizes viral infection.

The article presents solid data to back the authors’ primary claims that chSTING promotes apoptosis. However, the mechanistic details remain unclear. In order to increase the impact of this article, the authors could consider further investigating the mechanism by which chSTING initiates apoptosis, specifically by further determining the role or IRF7 in promoting apoptosis. For example, are IRF7-stimulated genes (ex., interferons, IFITs, etc.) necessary for chSTING/IRF7-mediated apoptosis, or does IRF7 directly interact with apoptotic signaling proteins? 

Major comments:

1. The authors should consider showing the effect of chSTING and mutants on apoptotic signaling (ex., PARP cleavage) and demonstrate that the DEVD and VAD inhibit these signals, similar to their analyses of autophagy in Fig. 4A.

2. The authors should consider explaining why they used IRF7+/- cells instead of IRF7 -/- cells. The authors should show the levels of IRF7 and phosphor-IRF7 via western blot. They could also perform siRNA-mediated knockdown to confirm results.

3. Fig. 5B. The results indicate that IRF7 is critical for the antiviral response to NDV. However, it is unclear based on their data whether IRF7 is functioning to promote chSTING-mediated apoptosis. The authors could strengthen these results by performing the following. First, quantify viral replication levels in this figure. Second, analyze apoptosis and apoptotic signaling in WT and IRF7+/- cells to determine if chSTING-mediated apoptosis is IRF7-dependent. Third, they should consider treating IRF7+/- with apoptosis inhibitors to determine if a further increase in viral replication is observed. If so, this would argue the apoptosis function of chSTING is separate from IR7-mediated gene induction.  Lastly, quantify the levels of phosphor-IRF7 in IRF7-/+ cells via both western blot and immunofluoresnce assays to determine the number of cells that display nuclear (phosphor) IRF7.

4. It is unclear if IRF7-mediated induction of antiviral genes leads to apoptosis. The authors could treat cells with actinomycin D to inhibit RNAP II to determine if gene induction is required for chSTING/IRF7-mediated apoptosis. 

Minor comments:

1. Grammatical errors in line 191 and line 192. Replace “shown” with “showed,” and “substitution” with “substituting.”

2. The descriptions in figure legends for figures 4D and 4E seem to be swapped as the current description for 4D mentions DEVD and VAD, which are shown in figure 4E. Likewise, 4E’s description mentions testing 3-MA and NH4Cl which are only shown in 4D.

Reviewer 2 Report

The authors in the present manuscript explore the role of Chicken STING protein and its role in antiviral responses. They explore the activity of the protein of interest in HEK293T cell line and chicken cell lines. The authors show that expression of chSTING in the HEK293T cell line along with cGAS was sufficient to show antiviral activity. They further posit that the mechanism of the antiviral activity is independent of IFN by generating the pLxSV domain substitutions and mutant S366A mutant which renders the down stream TBK - IF3 phosphorylation and IFN transcription inactive. The authors also posit that apoptosis is the mechanism for the antiviral response rather than autophagy.

The authors present the evidence in the form of microscopy images, western blots, qPCR, Flow cytometry etc., in support of their hypothesis. 

Major changes that need to be made is the the way the panels are presented, the Fluorescent images and western blots need to be quantified. Please provide the raw images for these type of data that will help us validate the changes in expression that are being claimed.

Line 20 and 21: "chSTING exerts antiviral activity in HEK293 cells and Chicken cells independent of IFN production. "

Line 21: Avoid " on the other hand".

Line 23: "  and apoptosis inhibitors rather than autophagy inhibitors antagonize"

Line 39: needs citation to the statement.

Line 61: define LC3 (Microtubule associated protein light chain 3). 

Methods:

Reagents: Be consistent with catalog numbers for the reagents used. I would prefer if the catalog numbers for all the kits and reagents are mentioned here, similar to the antibodies that were identified via their catalog numbers.

Line: 25: Please provide methods for the culture methods for the mentioned viruses in the supplementary methods. 

Line 129: what is the source of the ISRE and ELAM reporter vectors (clontech or other vendors or if in house generated material?)

Western blotting: How were the western blots analyzed or quantified? How many independent experiments were performed for the westerns? Please provide Raw images of the western blots. 

Do not cut - copy - paste pieces of western blot and assemble as a panel, instead repeat the blot with treatments in one continuous/contiguous blots. (figures2D, 4A).

Plaque assays: How were the images collected and analyzed? How was operator bias addressed?

Flowcytometry: Mention the Flow cytometer instrument.

Fluorescent Cell Imaging: Missing methods section, please describe how the fluorescent images of the virus infected cells were collected? Instrument used etc. Were they imaged with fixed settings to keep data collection consistent.

Results:

Line 191: Needs citation to your previous work.

Line 192: " substitution of the pLxVS motif with that of the porcine" 

Line 198: " These results indicate/suggest the IFN independent antiviral activity of"

Figure 1:

Please use larger image size for the Fluorescent microscopy images, quantify them via ImageJ software that is freely available. 

For the panel separate and label the western blots, the plaque images and bar graph as B, C, D. etc and the same for rest of the panel.

Western blots: Please quantify the western blots via densitometry analysis. Show the expression levels of cGAS and ch STING proteins. Normalize the GFP/RFP data to the ch STING protein level.  While the Actin control shows equal protein loading, showing evidence for cGAS and chSTING expression levels is critical to provide evidence thathe effect is indeed due to chSTING.

Missing: cGAS only control, N of three or three independent experiments are  a minimum claim an effect and to identify the experimental variability.

Figure 2:

D: The figure description says that this western blot is a non contiguous parts of the gel. Please repeat this western blot with the described treatments as a contiguous/continuous blot. Copy - paste editing/ assembling of western blot fragments is not acceptable.

Similar to figure1 please use larger images for microscopy, quantitate the western blots, normalized to chSTING or STINGâ–³CTT/Actin etc, pTBK/TBK etc. Actin protein levels show variability, so I would advise to normalize the western blots to a total protein stain. 

Lines 243, 244: Missing text??

Line 247 to 255 : Here your interpretation is that in chickens cells, the chSTING can function independently of cGAS, while in HEK cells cGAS was co-transfected with chSTING. Please show that the mamalian cell line either requires cGAS or that chSTING can function independently of cGAS in the HEK293 cell line. 

Figure 3: same suggestions for the microscopy images and western blots as in figures 1 and 2 quantitate and normalize the microscopy images and westerns.

Figure3G western Blot the magnitude of wt and pLxVS STING expression is weak compared to the mutants. Was this the case with other cell lines as well? If so, then is the antiviral activity dependent on the level of STING protein level, were these assays performed in prior work or by others? If so please cite them.

Figure4:

A) Again non-contiguous portions of the western blots appears to be re-arranged/ manipulated ? Please avoid that and repeat the blots and have the conditions in a contiguous manner and quantify.

Figure5: 

In  panel 5A in the western blots the actin loading control appears to show doublet bands suggesting modification of the actin protein via cellular processes. This was not observed in other blots. To avoid this phenomenon where the house keeping control proteins are affected, please use total protein staining for quantification.

Discussion:

Lines 370, 371: " chSTING pLxSV sub S366A inreased the NFkB signaling while CHSTING CTT reduced the signaling activity, but the antiviral activity remained similar"  or Please re word to make it easy to read and understand

Moderate level of grammatical and sentence structure needs to be changed to make it more enjoyable to read. I have identified certain sentences which could be written more clearly and concisely. However, I recommend the authors use an English Editor with background in Biology to improve the sentence structure. Over all very few grammatical errors and spelling mistakes were observed.
